# Impact of basic network motifs on the collective response to perturbations

Xiaoge Bao [1,2,3,12], Qitong Hu [1,4,12], Peng Ji [1,2,3] ✉, Wei Lin [2,3,5,6,7], Jürgen Kurths [3,8,9] & Jan Nagler [10,11] ✉

Many collective phenomena such as epidemic spreading and cascading failures in socioeconomic systems on networks are caused by perturbations of the dynamics. How perturbations propagate through networks, impact and disrupt their functions may depend on the network, the type and location of the perturbation as well as the spreading dynamics. Previous work has analyzed the retardation effects of the nodes along the propagation paths, suggesting a few transient propagation "scaling" regimes as a function of the nodes' degree, but regardless of motifs such as triangles. Yet, empirical networks consist of motifs enabling the proper functioning of the system. Here, we show that basic motifs along the propagation path jointly determine the previously proposed scaling regimes of distance-limited propagation and degree-limited propagation, or even cease their existence. Our results suggest a radical departure from these scaling regimes and provide a deeper understanding of the interplay of self-dynamics, interaction dynamics, and topological properties.

Signal propagation enables the proper functioning of complex systems on all natural and technological scales. Biochemical reaction networks underly signaling in cellular processes[1]. Other examples of collective phenomena include neural spike dynamics[2], gene regulatory dynamics[3–7], and epidemic spreading of infectious diseases, opinions or information[8–11]. Networked dynamical systems have proven suitable models for analyzing spatiotemporal signal spreading[12,13]. Yet, disentangling the effects resulting from the underlying structure and the collective dynamics on the networks remained conceptually difficult[14].

Thus, it came as a surprise when Hens and colleagues recently proposed universal features in signal propagation on networks[15], arising from studying the consequences from small irreversible perturbations of single units. They proposed a few markedly distinct types of propagation patterns that are determined topologically by the combination of the average number of nodes and their average degree along the propagation paths, resulting in a few fundamental asymptotic "scaling" regimes based on the nodes' degree but irrespective of features of motifs such as triangles. The majority of empirical networks, however, consist of motifs[16].

Network motifs are subgraphs, which can be acyclic or cyclic, directed or undirected, and play an important role in the design and evolution of complex networks[16]. Different $n$-node subgraphs account for elementary computational circuits and play various functional roles in information procession[17], including 13 types of directed three-node subgraphs[16,17]. Three-nodes motifs in particular may enhance the resilience to perturbations in power grids[18,19], and play a central role in the emergence and maintenance of social networks[20,21]. A number of candidate motifs may serve as basic but functionally important building blocks of regulatory and transcription networks[22–25]. Research on the extent and function of motifs has provided a better

[1]Institute of Science and Technology for Brain-Inspired Intelligence, Fudan University, Shanghai, China. [2]Key Laboratory of Computational Neuroscience and Brain-Inspired Intelligence (Fudan University), Ministry of Education, Shanghai, China. [3]Research Institute of Intelligent Complex Systems and MOE Frontiers Center for Brain Science, Fudan University, Shanghai, China. [4]School of Mathematical Sciences, Shanghai Jiao Tong University, Shanghai, China. [5]School of Mathematical Sciences, SCMS, SCAM, and CCSB, Fudan University, Shanghai, China. [6]State Key Laboratory of Medical Neurobiology, Institutes of Brain Science, Fudan University, Shanghai, China. [7]Shanghai Artificial Intelligence Laboratory, Shanghai, China. [8]Potsdam Institute for Climate Impact Research, Potsdam, Germany. [9]Humboldt University, Berlin, Germany. [10]Deep Dynamics, Frankfurt School of Finance & Management, Frankfurt, Germany. [11]Centre for Human and Machine Intelligence, Frankfurt School of Finance & Management, Frankfurt, Germany. [12]These authors contributed equally: Xiaoge Bao, Qitong Hu. ✉e-mail: pengji@fudan.edu.cn; jan.nagler@gmail.com

understanding of the complex operational dependencies between nodes[23–28].

Here, we study how basic undirected motifs, in particular edges and triangles, determine the response times to perturbations – as a function of the network structure and its dynamics. This allows us to identify genuine scaling regimes jointly arising from nodes' degree and motifs. Our framework of response dynamics to perturbations on networks conceptually links the small and large scale topology of a network with the spatiotemporal spreading induced by a single small perturbation. In particular, we identify and predict the impact and interplay of propagation paths, degree and motif distributions, and interaction dynamics.

## Results

### Model

We characterize the network dynamics by pairwise interacting nodes,

$$\dot{x}_i(t) = F\big(x_i(t)\big) + \sum_{j=1}^{N} A_{ij} H_1\big(x_i(t)\big) H_2\big(x_j(t)\big), \tag{1}$$

which describes the evolution of the state variables $x_i$ of node $i$, where throughout the manuscript $i = 1...N$. Different dynamics on networks are captured by the triplet $\{F(x_i), H_1(x_i), H_2(x_j)\}$ comprising of (possibly) nonlinear functions, where $F(x_i)$ specifies the self-dynamics governing influx, leaking dynamics, degradation or reproduction[29–34]. The terms $H_1(x_i)$ and $H_2(x_j)$ determine the the adjacent interactions of node $i$ with its neighbors, such as infection, mutualism and competition[29–34]. The connectivity matrix $A$ accounts for the connections.

The unperturbed system is considered in a stationary collective state, which we characterize by the set of $N$ stable equilibria, $x_i^* = x_i(t = 0)$. We examine the signal propagation by studying the transient dynamics induced by a permanent perturbation $\Delta x_m$ on the steady state $x_m^*$ of the source node $m$. The perturbation forces nodes to

transition to the shifted states $x_i(\infty) = x_i^* + \Delta x_i(\infty)$. For each node $i$, we characterize this transition period in terms of the response time $\tau_{im}$, defined as the time the response ratio

$$\delta_i(t) = \frac{\Delta x_i(t)}{\Delta x_i(\infty)} \tag{2}$$

takes the fixed value $\delta_i(\tau_{im}) = \eta$.

The evolution of the collective dynamics of all states may exhibit a variety of intricate spatiotemporal patterns, which we quantify by the relationship between the response time $\tau_{im}$ and the node degree $d_i$, the number of node $i$'s edges. In contrast to previous work, our framework accounts for multipath connections between source and target, which are salient features of empirical networks. Figure 1 illustrates the main mechanisms underlying the impact of motifs on response times in a protein-protein network[35], a small-world network, and an Erdös-Rényi network[11]. Basic motifs, defined as convex regular $n$-gons such as edges ($n = 2$), triangles ($n = 3$), squares ($n = 4$), and pentagons ($n = 5$) are ubiquitous units of random networks, as shown in Fig. 1b, c. However, as detailed in Fig. 1c, single edges disjoint from motifs are comparably rare, leading triangles to dominate networks.

To demonstrate the dynamical role of triangles we study population dynamics on networks and quantify the relative response to a perturbation $\frac{\Delta x_i(t)}{\Delta x_i(\infty)}$ with respect to different number of triangles. As shown in Fig. 1d, target nodes $i$ as part of edges respond faster than target nodes as part of triangles, hence impacting the response time qualitatively. As shown in Fig. 1e, not only the response time may depend on the number of triangles along a path, but also the response time may vary, depending on the local network structure, even for the same number of triangles along the path. This demonstrates the impact basic motifs may have on local response dynamics on networks.

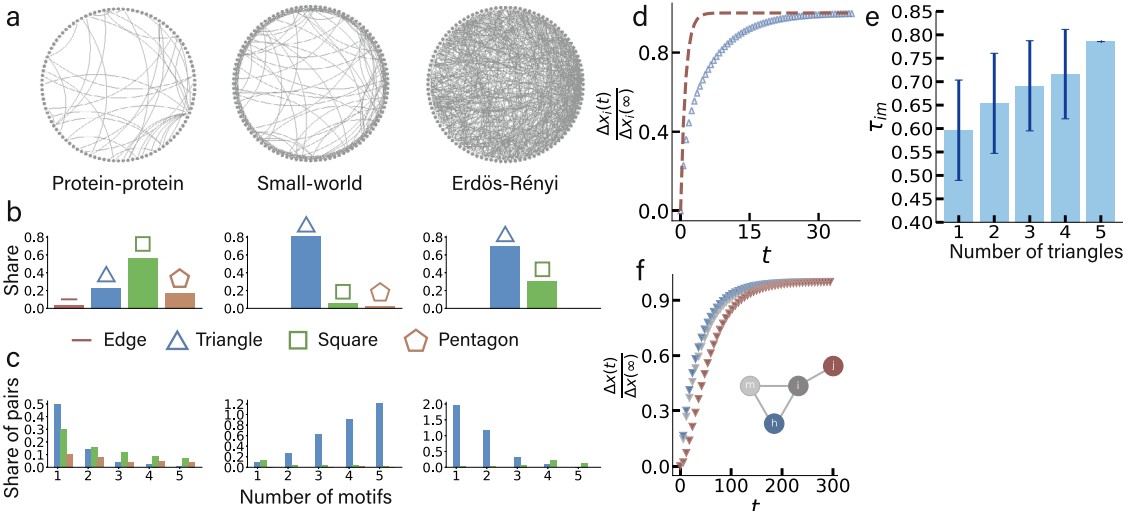

**Fig. 1 | Prevalence and dy5gnamical impact of motifs in random networks.**
**a** From left to right: Schematic plot of protein-protein network (shown for $N = 100$ nodes from total $N = 2035$)[35], small-world network, and Erdös-Rényi (ER) network with network size $N = 100$ and average degree 10. **b** Share of motifs (edges, triangles, squares, pentagons) for a network. Triangles play important roles, no matter for the networks with dense or random connections, or highly clustered networks. **c** Share of edges as part of basic motifs (triangles, squares, pentagons). The share is calculated by the number of edges as part of corresponding motif divided by the network size. Most edges form triangles, and for the small-world network with high clustering more and more edges form larger number of triangles, demonstrating the dominance of triangles. **d** Target nodes $i$ as part of the (independent) edges (brown) respond faster than other target nodes $i$ as part of triangles (blue), as

shown by $\frac{\Delta x_i(t)}{\Delta x_i(\infty)}$. **e** Histogram of average propagation time from randomly chosen sources to their randomly chosen adjacent target nodes, as a function of the number of triangles (1–5). Error bars indicate standard deviation. **d** and **e** are based on single ER network realizations with the linking probability $p = 0.10$ and the network size $N = 100$ (rightmost network in (**a**)). **f** Perturbation response $\frac{\Delta x_i(t)}{\Delta x_i(\infty)}$ of nodes $i, j$ and $h$ of the four-node network model as shown inside the panel, with target node $i$ and its neighbors $j$ and $h$, to a perturbation on the source $m$. Responses of $j$ and $h$ differ, mainly due to their different positions relative to $m$, but all nodes respond non-instantaneously on the same time scale, which motivated us to derive the framework, see main text. Node $j$ responds the slowest because it is two edges apart from source $m$. For (**d**), (**e**) and (**f**), the system is governed by population dynamics $\dot{x}_i(t) = -Bx_i^a + \alpha \sum_{j=1}^{N} A_{ij} x_j^b$, where $B = \alpha = 0.01$, $a = 1.2$, $b = 1.1$.

## Local propagation

To systematically quantify the impact of basic motifs on response dynamics on networks, we formulate a general theoretical framework based on Eq. (1). This requires the determination of the responses $\Delta x_i(t)$ to the perturbation $\Delta x_m$. We are first concerned with the quantification of the local propagation from node $m$ to its neighbor $i$. For small perturbations, employment of linear response theory allows us to formulate the response dynamics,

$$\Delta \dot{x}_i(t) = -\frac{1}{J_i}\Delta x_i(t) + H_1(x_i^*)\sum_{j\neq m}^{N} A_{ij}H_2'(x_j^*)\Delta x_j(t) + A_{im}H_1(x_i^*)H_2'(x_m^*)\Delta x_m,$$
(3)

where $H_2'(x)$ represents the derivative $dH_2(x)/dx$, which is evaluated at the initial states $x_j^*$ and $x_m^*$, while $J_i$ represents the self-dynamics of the form

$$J_i = -1 \left/ \left( H_1(x_i^*)\left[\frac{F(x_i^*)}{H_1(x_i^*)}\right]' \right).\right.$$
(4)

The right-hand side of Eq. (3) holds three contributions to the response: self-dynamics, a sum specifying the adjacent interaction dynamics $j \neq m$, and the response of node $i$ directly induced by source $m$. The perturbed system converges for $t \to \infty$ to the system's new collective stationary state, with responses $\Delta x_i(\infty)$. We quantify the responses in finite time by the ratio $\delta_i(t)$, Eq. (2), whose solutions are obtained by reformulating the linear response equation (3),

$$\ln(1 - \delta_i(t)) = -\frac{1}{J_i}\int_0^t (1 - \mathcal{E}_{im}(\tau))d\tau,$$
(5)

where $\mathcal{E}_{im}(t)$ represents the contribution from the neighbors' (adjacent) dynamics and is defined as

$$\mathcal{E}_{im}(t) = J_i H_1(x_i^*)\sum_{j\neq m}^{N} A_{ij}H_2'(x_j^*)\frac{\Delta x_j(t) - \Delta x_j(\infty)}{\Delta x_i(t) - \Delta x_i(\infty)}$$
(6)

and the response times are determined by the solutions $\delta_i(t = \tau_{im}) = \eta$.

If we assumed that the states of adjacent nodes jump instantly to their new stationary state, $\Delta x_j(\tau_{im}) \approx \Delta x_j(\infty)$, the term $\mathcal{E}_{im}(t)$ would vanish, $\mathcal{E}_{im}(t) \approx 0$. Previous work[15,36] that has hypothesized three distinct spatiotemporal scaling regimes has implicitly assumed this premise. However, it is important to emphasize that the premise is not always valid, especially for networks with prevalent motifs, and not even for the four-node network in Fig. 1f. To recognize this, we focus on this four-node network and its responses to a perturbation on node $m$, $\Delta x_i(t)$, $\Delta x_h(t)$ and $\Delta x_j(t)$, where nodes $m$, $i$ and $h$ form a triangle, and node $j$ is adjacent to node $i$. Note that edge $(i$–$j)$ is referred to as an *independent edge* as it is not directly connected to the source $m$. The convergence behaviors of the responses $\Delta x_h(t)$ and $\Delta x_j(t)$ are comparable with $\Delta x_i(t)$, which is conflicting to a small $\mathcal{E}_{im}(t)$. We also observe that $\Delta x_h(t)$ and $\Delta x_j(t)$ exhibit qualitatively different asymptotic behaviors. Also note that the response of node $j$ is slower than those of nodes $h$ or $i$ because $j$ is two edges apart from source $m$, which has a stronger effect than the overall faster responses of nodes as part of independent edges compared to triangles.

Apart from triangles, other motifs may also occupy a large proportion of the networks, as shown in Fig. 1b, c. To further analyze the impact of triangles and other basic motifs, we compare the response times in small synthetic networks with and without localized signal flow disruptions (see Supplementary Material, Tables S1 & S2). We study $n$-gons, from triangles ($n = 3$) to pentagons ($n = 5$) finding that differences of the response time are larger for the smaller $n$. This means that the most significant impact on the response time can be attributed to independent edges and triangles, which prompt us to

decompose networks into independent edges, that is, 2-gons such as the $(i$–$j)$-edge in the example in Fig. 1f, and triangles (3-gons).

We quantify the response times from three basic perspectives: self-dynamics, independent edges and triangles. The unperturbed system is considered in a steady state, characterized by $N$ stable equilibria $x_i^*$. The relationship between the intrinsic dynamics of node $i$ and its adjacent dynamics in the steady state follows immediately from Eq. (1) and $\dot{x}_i = 0$,

$$\frac{F(x_i^*)}{H_1(x_i^*)} = -\sum_{j=1}^{N} A_{ij}H_2(x_j^*).$$
(7)

Averaging allows us to compute the contribution of adjacent dynamics in the mean-field

$$\overline{\mathcal{H}} := \frac{1}{N}\sum_{i=1}^{N}\frac{1}{d_i}\sum_{j=1}^{N} A_{ij}H_2(x_j^*),$$
(8)

which is then used to simplify the relationship (7) as $R(x_i^*) := -\frac{F(x_i^*)}{H_1(x_i^*)} \approx d_i \times \overline{\mathcal{H}}$, in which the degree serves as a coupling constant of the intrinsic dynamics of node $i$ and its adjacent connections. In the unperturbed steady system, the equilibria $x_i^*$ can be expressed as the inverse function of $R(x_i^*)$ with respect to the degree $d_i$,

$$x_i^* = R^{-1}(d_i\overline{\mathcal{H}}).$$
(9)

Combining Eqs. (5) and (9), we derive the response time $\tau_{im}$ of node $i$ as a function of its degree $d_i$, as

$$\tau_{im} = \frac{-J_i\ln(1 - \eta)}{1 + \frac{1}{\ln(1-\eta)}\frac{\eta}{1-\eta}\mathcal{E}_{im}},$$
(10)

where $\mathcal{E}_{im} \approx d_i Q_i \overline{Q_{im}}$ results from the node $i$'s intrinsic dynamics, through $Q_i = J_i(x_i^*)H_1(x_i^*)H_2'(x_i^*)$ and its adjacent nodes' mean dynamics, through $\overline{Q_{im}}$, see the Supplementary Material. Since $x_i^* = R^{-1}(d_i\overline{\mathcal{H}})$, the quantity $Q_i$ is a function of degree $d_i$, while the mean-field quantity $\overline{Q_{im}}$ is independent of the degree. In the Supplementary Material, the Hahn expansion of $Q_i$ leads to its leading power, $Q_i \sim d_i^{\Pi_Q(\infty)}$, with the constant $\Pi_Q(\infty)$. In the large-degree limit, $d_i \to \infty$, we obtain

$$\mathcal{E}_{im} \sim d_i^{\theta_Q},$$
(11)

with the scaling exponent $\theta_Q = \Pi_Q(\infty) + 1$. The exponent $\theta_Q$ is determined by the intrinsic dynamics but is independent of $d_i$.

In the large-degree limit, for $\theta_Q < 0$ the contributions to $\tau_{im}$ from adjacent nodes vanish such that the response time becomes independent of $m$ and is well approximated by

$$\tau_i = -J_i\ln(1 - \eta).$$
(12)

This result coincides with existing literature and mechanistically explains the validity of previous theoretical results for a number of dynamics in the large-degree limit[15,36] – although the adjacent interactions as quantified by $\mathcal{E}_{im}$ are not considered.

The self-dynamics term $J_i$ can be expanded as a function of the degree $d_i$, as $J_i \sim d_i^{\theta_J}$, where the scaling exponent $\theta_J$ is the leading power of the Hahn expansion. In doing so, we find that the response time $\tau_i$ exhibits the scaling relation

$$\tau_i \sim d_i^{\theta_J},$$
(13)

where the scaling exponent $\theta_J$ is determined by the intrinsic dynamics but independent of adjacent connections. The scaling relationship (13) highlights the contribution of both the structural features and system dynamics on the response time, yet it is a disentanglement of self-

dynamics $J_i$ and degree $d_i$. In that way, this finding is in agreement with previous work on three distinctive dynamic regimes[15].

In contrast, for $\theta_Q > 0$, the contributions from adjacent nodes may be substantial, even for large degree. In this case, both the self-dynamics and the adjacent dynamics contribute to the response time, remarkably exhibiting a scaling relation

$$\tau_i \sim d_i^{\theta_J - \theta_Q}, \tag{14}$$

where the scaling is affected by the adjacent dynamics, through the exponent $\theta_Q$.

It is important to relate the exponent $\theta_Q$ to prototypical network dynamics. As derived in the Supplementary Material, we find that regulatory, human, mutualistic, biochemical and epidemics dynamics are characterized by $\theta_Q < 0$, whereas population and inhibitory neuronal dynamics show $\theta_Q > 0$, as shown in Table 1. As the model parameters are assumed to be non-negative but otherwise arbitrary, Table 1 characterizes a substantial range of dynamical systems. The theoretical derivation, Eq. (10), exactly predicts the response time $\tau_i$. The scaling relationships, Eqs. (13) and (14), characterize the interplay between network topology, the self-dynamics and its adjacent dynamics in the asymptotic regime $d_i \to \infty$. We find that both the theoretical derivation and the scaling predictions are in good agreement with simulations for regulatory and population dynamics, as supported by Fig. 2a–e. For regulatory dynamics, characterized by $\theta_Q < 0$, the response time $\tau_i$ is

**Table 1 | Scaling exponents $\theta_J$ and $\theta_Q$ for prototypical dynamical models**

| Model | Dynamical Equation | $\theta_J$ | $\theta_Q$ |
|---|---|---|---|
| Regulatory ($\mathbb{R}$) | $\dot{x}_i(t) = -Bx_i^a(t) + \alpha \sum_{j=1}^N A_{ji} \frac{x_j^b(t)}{1+x_j^b(t)}$ | $\frac{1}{a}-1$ | $-\frac{b}{a}$ |
| Human ($\mathbb{H}$) | $\dot{x}_i(t) = -Bx_i^{a+b}(t) + \alpha x_i^b(t) \sum_{j=1}^N A_{ji}\left(y_0 - x_j^{-c}(t)\right)$ | $\frac{1-b}{a}-1$ | $-\frac{c}{a}$ |
| Epidemics ($\mathbb{E}$) | $\dot{x}_i(t) = -Bx_i(t) + \alpha(1-x_i(t)) \sum_{j=1}^N A_{ji}x_j(t)$ | $-1$ | $-1$ |
| Mutualistic ($\mathbb{M}$) | $\dot{x}_i(t) = Bx_i(t)\left(1 - \frac{x_i^a(t)}{C}\right) + \alpha x_i(t) \sum_{j=1}^N A_{ji}\frac{x_j(t)}{1+x_j(t)}$ | $-1$ | $-\frac{1}{a}$ |
| Population ($\mathbb{P}$) | $\dot{x}_i(t) = -Bx_i^a(t) + \alpha \sum_{j=1}^N A_{ji}x_j^b(t)$ | $\frac{1}{a}-1$ | $\frac{b}{a}$ |
| Biochemical ($\mathbb{B}$) | $\dot{x}_i(t) = B - Cx_i(t) - \alpha x_i(t) \sum_{j=1}^N A_{ji}x_j(t)$ | $-1$ | $-1$ |
| Inhibitory ($\mathbb{I}$) | $\dot{x}_i(t) = -Bx_i(t)\left(1 - \frac{x_i(t)}{C}\right)^2 + \alpha x_i(t) \sum_{j=1}^N A_{ji}x_j(t)$ | $-1$ | $\frac{1}{2}$ |

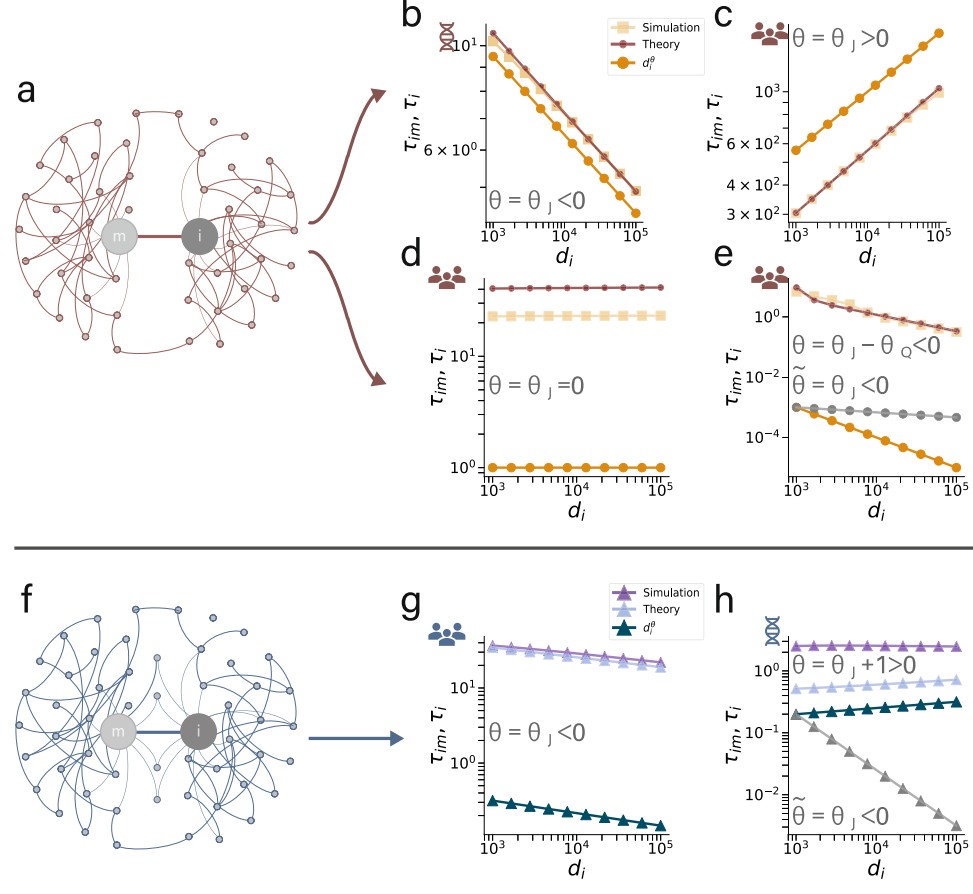

**Fig. 2 | Response time scaling for local propagation. a** Network with perturbation at $m$ with target node $i$ having $d_i$ edges (referred to as independent edges). **b** Propagation time $\tau_i$ (time from source $m$ to $i$) as a function of degree $d_i$ for network (**a**) with regulatory dynamics ($\theta_J = \frac{1}{a}-1$ and $\theta_Q = -\frac{b}{a} < 0$) and scaling exponent $\theta = \theta_J$. **c** Propagation time $\tau_i$ for network (**a**) with population dynamics ($\theta_J = \frac{1}{a}-1$ and $\theta_Q = \frac{b}{a} > 0$) and $\theta = \theta_J > 0$. **d** Propagation time $\tau_i$ for network (**a**) for population dynamics ($\theta = \theta_J = 0$). **e** Propagation time $\tau_i$ for network (**a**) ($\theta = \theta_J - \theta_Q < 0$). Theory, Eq. (10), and scaling relationship, Eqs. (13) and (14), in comparison with simulation. According to Eq. (10), we predict $\tau_i \sim d_i^{\theta_J-\theta_Q}$, (14), which is in good agreement with the simulated response time but disagrees with the prediction from existing literature $\tau_i \sim d_i^{\tilde{\theta}} = d_i^{\theta_J}$ (gray). **f** Network with perturbation at source $m$ and target at node $i$ having $d_i$ triangles attached. **g** Propagation time $\tau_i$ for network (**f**) with population dynamics ($\theta_J = \frac{1}{a}-1$ and $f < 1$) and the scaling exponent $\theta = \theta_J$. The scaling prediction according to Eq. (15), $\tau_i \sim d_i^{\theta_J}$, is in agreement with numerics. **h** Propagation time $\tau_i$ versus $d_i$ for network (**f**) with regulatory dynamics ($\theta_J = \frac{1}{a}-1$ and $f \approx 1$) and $\theta = \theta_J + 1$. According to Eq. (16), scaling is predicted as $\tau_i \sim d_i^{\theta_J+1}$, which is close to the simulation response time scaling but in disagreement with the prediction from existing literature, $\tau_i \sim d_i^{\tilde{\theta}} = d_i^{\theta_J}$. **b** and **h**, regulatory dynamics, $\dot{x}_i(t) = -Bx_i^a + \alpha \sum_{j=1}^N A_{ij}\frac{x_j^b}{1+x_j^b}$, where $a = 1.2, b = 2.0$ for (**b**), and $a = 10.0, b = 2.0$ for (**h**). **c**, **d**, **e** and **g**, population dynamics $\dot{x}_i(t) = -Bx_i^a + \alpha \sum_{j=1}^N A_{ij}x_j^b$, with $a = 1.2, b = 1.0$ for (**c**), and $a = 1.0, b = 0.2$ for (**d**), and $a = 1.2, b = 0.6$ for (**e**), and $a = 1.2, b = 0.5$ for (**g**), and $B = \alpha = 0.01$.

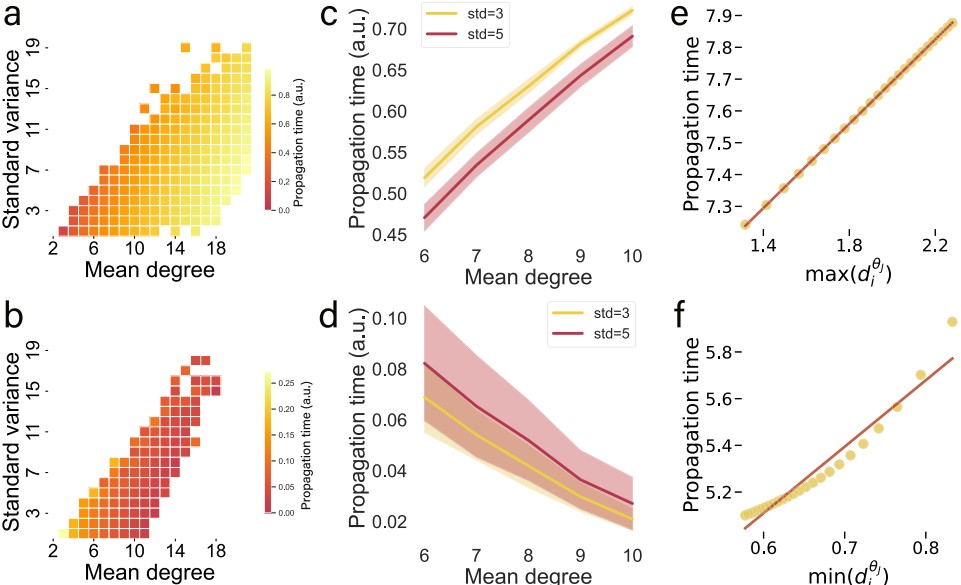

**Fig. 3 | Impact of degree sequences on global propagation. a, b** Propagation time as a function of average degree and degree variance for a chain of 8 nodes with random degree sequence and the first node set as a perturbation source. Propagation time is averaged across different realizations. **c, d** Average propagation time versus mean degree, for sequences of fixed variance 3 and 5, respectively. Response time and mean degree in (**c**) show positive correlation, while (**d**) shows a negative correlation. Response time versus $\max(d_i^{\theta_J})$ for $\theta_J = 0.25 > 0$ (**e**) and $\min(d_i^{\theta_J})$ for $\theta_J = -0.17 < 0$ (**f**). The linear relationship (red fitting line) indicate that the simulated response time is well described by $\max(d_i^{\theta_J})$ or $\min(d_i^{\theta_J})$. **a–f** Regulatory dynamics, $\dot{x}_i(t) = -Bx_i^a + \alpha \sum_{j=1}^N A_{ij} \frac{x_j^b}{1+x_j^b}$, where $a = 0.8$; $b = 0.5$ (**a, c, e**), $a = 1.2$; $b = 2.0$ (**b, d, f**), and $B = \alpha = 0.01$. **a–d** node degrees are sampled randomly. **e–f** the fifths node's degree, $d_5$, is varied from 5 to 30, while all other node degrees are kept fixed, $d_{i \neq 5} = 2$.

determined solely by the self-dynamics, governed by the scaling relationship Eq. (13), as shown in Fig. 2b. For population dynamics, characterized by $\theta_Q > 0$, we find three distinctive dynamic regimes. In the degree-limited regime with $\theta_J > 0$ in Fig. 2c and the distance-limited regime with $\theta_J = 0$ in Fig. 2d, the term $\mathcal{E}_{im}$ is negligible due to the degree limitation, and scaling is only determined by the self-dynamics Eq. (13). However, in the composite dynamic regime ($\theta_J < 0$), $\tau_i$ is determined by both the self-dynamics and the adjacent dynamics as predicted by Eq. (14) and supported by Fig. 2e. While existing literature disregarded the effects from adjacent dynamics and predicts only $\tilde{\theta} = \theta_J$[15], leading to inaccurate scaling as shown in Fig. 2e, our prediction, $\theta = \theta_J - \theta_Q$, is asymptotically exact. Taken together, we observe scaling of the response time that depend on both the self-dynamics and the adjacent dynamics, even for local tree-like networks. But how relevant is this composite dynamic regime?

For a triangle-dominated topology as shown in Fig. 2f, we derive the response time $\tau_i$ from Eqs. (5) and (9) as

$$\tau_i = -\ln(1-\eta)J_i \frac{1+\mathcal{C}_{im}}{1+(1-f)\mathcal{C}_{im}}, \tag{15}$$

where $\mathcal{C}_{im}$ can be approximated by the product of the degree-independent mean-field term $\overline{Q_{im}}$ and $d_i$, $\mathcal{C}_{im} \approx d_i \overline{Q_{im}}$. Note that $f$ is a constant that depends on the system dynamics, but not on $d_i$. Now, for $f \ll 1$ and a large degree, $\mathcal{C}_{im}$ drops out in Eq. (15). In this case, triangles have a negligible impact on the scaling and the scaling relation is well approximated by $\tau_i \sim d_i^{\theta_J}$, as shown in Fig. 2g. For $f \approx 1$, the presence of triangles, as shown in Fig. 2h, results in the extra factor $d_i$ resulting from $\mathcal{C}_{im}$ such that

$$\tau_i \sim d_i^{\theta_J + 1}. \tag{16}$$

As shown in Fig. 2f–h for regulatory and population dynamics, both predictions, Eqs. (15) and (16) are well confirmed by our computer simulations. This establishes that basic motifs may crucially determine scaling, even in simple networks.

In the Supplementary Material, we derive the respective explicit solutions of the response time, not only for the asymptotic scaling regime but also for the regime of small degrees, which allows us to fully characterize the scaling regimes. In addition to regulatory and population dynamics, we develop the system dynamics framework for human[29], epidemic[30], mutualistic[31], biochemical and inhibitory dynamics[32–34,37,38] (see Table 1). Our results consistently quantify the impact of basic motifs on local response dynamics on networks.

## Global propagation

Thus far, we have established a versatile theoretical framework for local signal propagation. To formulate a framework for global signal propagation, it is helpful to examine a source-centric representation, which allows to theoretically track all possible propagation paths from the perturbed central node to all distant nodes. Specifically, we impose a layer-to-layer topology with the source node in the center and study the interplay of the intrinsic and adjacent dynamics, employing our developed framework for the local propagation as a building block. Denote by $T(m \rightarrow i_k)$ the propagation time from node $m$ to the $i_k$-th node through a pathway, specifying when the signal response ratio $\frac{\Delta x_{i_k}(t)}{\Delta x_{i_k}(\infty)}$ attains the threshold $\eta$. The signal propagation is expressed layer-wise in terms of $T(m \rightarrow i_k)$, which we compute recursively,

$$T(m \rightarrow i_k) = T(m \rightarrow i_{k-1}) - \frac{\Delta x_{i_k}(T(m \rightarrow i_{k-1})) - \eta \Delta x_{i_k}(\infty)}{\Delta \dot{x}_{i_k}(T(m \rightarrow i_{k-1})) - \eta \Delta \dot{x}_{i_k}(\infty)} \tag{17}$$

where the subscript $k$ of index $i_k$ stands for the distance (the number of edges along the path) to the perturbed node $m$.

Based on the Gauss Iterative Method combined with the generalization of Taylor expansion[39], we solve Eq. (17), yielding

$$T(m \rightarrow i_k) \sim \mathbf{g}^T \mathbf{D}, \tag{18}$$

with vector $\mathbf{g}^T = (g(1) g(2) \ldots g(k))$, whose components are monotonic decreasing functions that approach 1 as $k \rightarrow \infty$. The components $g(h)$ of

**g** are approximately proportional to a product,

$$g(h) \sim \prod_{j=h+1}^{k} \left( \frac{1}{1 - \mathcal{E}_{i_j i_{j-1}}} \right), \tag{19}$$

accounting for the accumulated dynamical effects from independent edges of layers higher than $h$. Note that the components may vary across different system dynamics, but are of the same order for a given dynamics (see Supplementary Material). The vector **D** depends on the degree sequence $d_{i_k}$ and the parameters characterizing the adjacent dynamics,

$$\mathbf{D} = \mathbf{D}\left( d_{i_1} \cdots d_{i_k} \right) = \left( -\ln(1-\eta) \frac{d_{i_1}^{\theta_J}}{1 - C_1 C_2 d_{i_1}^{\theta_Q}} \frac{d_{i_2}^{\theta_J}}{1 - C_2 d_{i_2}^{\theta_Q}} \cdots \frac{d_{i_k}^{\theta_J}}{1 - C_2 d_{i_k}^{\theta_Q}} \right)^{\mathsf{T}}, \tag{20}$$

where $C_1$ and $C_2$ are constants, and the exponent $\theta_J$ characterizes the self-dynamics, while $\theta_Q$ depends also on the adjacent dynamics, see Table 1. The scalar product (18) is the summation of the propagation times through the layers, where the vector **g** weights the degree sequence along the pathway as held in **D**. Because the components of **g** are given as product (19), degree fluctuations do not average out as it would for an additive structure. This is why and how the degree sequence affects the global propagation time, $T(m \rightarrow i_k)$.

Taken together, equation (18) predicts that not only the average degree, as assumed in previous literature[15], but also the variance and even the degree sequence along the propagation path may crucially determine the response times to a perturbation. To further illustrate this, we study how signal propagation is impacted by the average degree and degree variance of the nodes forming a chain. Figure 3a, c show for regulatory dynamics with $\theta_J > 0$ that the propagation time increases with increasing mean degree, and decreases with increasing standard variation of the randomly chosen degree sequence. In contrast, for $\theta_J < 0$, the propagation time decreases with the mean degree and the standard variation, as shown in Fig. 3b, d.

As we expect the global signal propagation $T(m \rightarrow i_k)$ to be sensitive to the degree sequence along the propagation path, in the Supplementary Material, we systematically study degree sequences with large standard variation, which leads us to two crucial observations. First, for $\theta_J > 0$, the propagation time is dominated by the largest node degree $\max(d_i^{\theta_J})$ along the chain, which is consistent with the numerics as shown in Fig. 3e. Second, for regulatory dynamics with $\theta_J < 0$, the propagation time is dominated by the smallest degree $\min(d_i^{\theta_J})$, which is supported by Fig. 3f.

To study the impact basic motifs have on empirical networks, we analyzed a protein-protein network[35], where we control the clustering coefficient by edge rewiring. The central node is the source node inducing a perturbation. Nodes in the same layer (with the same radial distance) have the same shortest path length to the source, as shown in Fig. 4a. Brown nodes indicate the arrival of the perturbation in (an arbitrarily) fixed time, for fixed $\eta$. For this dynamics, triangles inhibit propagation. The average propagation time increases as a function of number of triangles and layers, as shown in Fig. 4b. Similarly, we find that signal propagation is increasingly inhibited with increasing clustering coefficient, as shown in Fig. 4c. For an extensive analysis of this empirical network, together with a detailed investigation for a number of prototypical networked systems, whose arbitrary parameter ranges are expected to characterize a wide range of empirical networked dynamical systems, we refer to the Supplementary Material.

## Discussion

Triangles and other loops have always limited the understanding of the interplay of function and structure in networks. We have developed

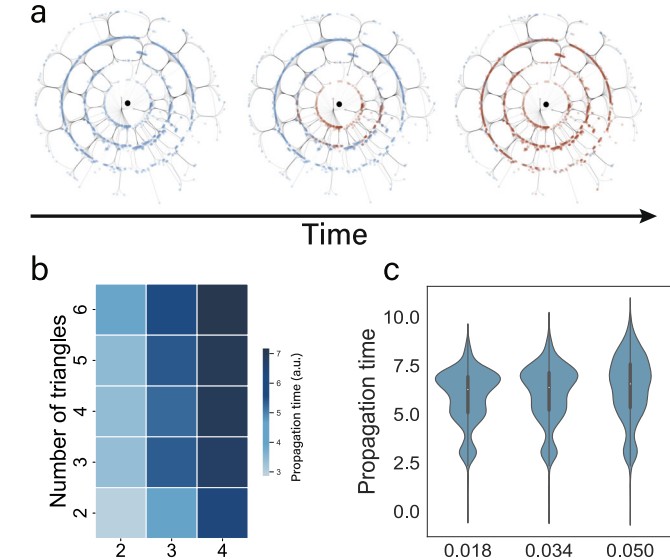

**Fig. 4 | Impact of triangles on global propagation in protein-protein networks.** **a** Signal propagation snapshots of protein-protein networks[35]. Central node is the source node inducing a perturbation. Nodes in the same layer (same radial distance) have same shortest path length to the source. Brown nodes indicate the arrival of the perturbation in (an arbitrarily) fixed time. **b** Average propagation time as a function of number of triangles and layers in (**a**). For a given layer, triangles slow propagation. Average propagation time increases with both number of triangles and layers. **c** Signal propagation in rewired networks with varying the clustering coefficient by edge rewiring. For all three networks, high clustering and triangle density slows the spreading of the perturbation across layers. Panels for regulatory dynamics, $\dot{x}_i(t) = -Bx_i^a + \alpha \sum_{j=1}^{N} A_{ij} \frac{x_j^b}{1 + x_j^b}$, with $B = \alpha = 1$, $a = 0.8$, $b = 0.5$, and $N = 2035$.

analytical tools that allow to capture the impact of simple undirected motifs on the system dynamics. Our developed framework not only helps disentangle joint effects but provides a deeper understanding of the interplay of self-dynamics, interaction dynamics, and topological properties. Our analysis suggests a radical departure from the previously proposed concepts of distance-limited propagation and degree-limited propagation. In distance-limited propagation the response time scaling is said to be dominated by the propagation path length but not by the edge density along the path. Vice versa, for degree-limited propagation the response time scaling is said to be dominated by the mean degree, but not the propagation path length. We have demonstrated here by independent methods that when the propagation is drastically slowed, or accelerated, that may not necessarily result from edges or hubs but from cycles, in particular triangles. Our analysis is based on a network decomposition into independent edges and edges as part of motifs. The developed framework predicts genuine scaling exponents, no matter if the propagation dynamics is dominated by hubs, by the path length, or by basic motifs. For paths with large average degree, as abundant in social and other empirical networks, the prediction of propagation time using asymptotic scaling as proposed by existing literature may be orders of magnitude off, as the scaling exponent may fall in an unrelated universality class. We have overcome this inconsistency by introducing two topology-independent exponents that quantify the universality class of the local response dynamics on networks.

Network motifs are abundant in synthetic and empirical networks and systematically impact the response dynamics to perturbations. We have provided a versatile toolbox that may not only help understanding response dynamics on networks but also provide the mathematical building blocks for extensions such as genuine universality classes for dynamics on directed multiplex networks. Yet, it is

important to note that the developed analytical tools are built on linear response theory to quantify the dynamics in the steady state as a linear response to a small permanent perturbation of a single unit. Large dynamic perturbations, for instance, may force the system to transition into steady states not predicted by linear response theory. Theoretical extensions covering this subject remain challenging but deserve future attention.

## Data availability

The data in this study is freely accessible at https://github.com/QitongHu2000/Impact-of-motifs-data.

## Code availability

Code for replicating this study is freely accessible at https://github.com/QitongHu2000/Impact-of-motifs-main.

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

## Acknowledgements

P.J. is supported by National Science and Technology Innovation 2030 Major Program (2021ZD0204500, 2021ZD0204504), National Natural Science Foundation of China (62076071), and Shanghai Municipal Science and Technology Major Project(2018SHZDZX01). W.L. is supported by National Natural Science Foundation of China (11925103), STCSM (22JC1402500), and Shanghai Municipal Science and Technology Major Project (2021SHZDZX0103). J.K. was supported by the Russian Ministry of Science and Education (Agreement No. 075-15-2020-808).

## Author contributions

P. J. and J.N. conceived the project, designed the study and wrote the manuscript. Q. H. designed initial simulations and developed the formalism. X.B. re-conducted and verified Q.H.'s work. J. K and W. L. contributed to writing the paper.

## Funding

## Competing interests

The authors declare no competing interests.
