## [Peer Review File · Nature Communications]

REVIEWERS' COMMENTS

Reviewer #1 (Remarks to the Author):

The manuscript "Impact of higher-order network motifs on response dynamics" focuses on the effects that the presence of triangles and other types of higher-order motifs have on the characteristic time over which a perturbation propagates through a network. The authors in particular focus on specific type of motifs, that is, closed paths of variables length (edge, triangle, square, pentagon, etc), and derive general expressions valid for many different types of models (ranging from epidemic to biochemical ones).

The central result is --at the level of local perturbation-- a direct link between the response time of a unit i with its degree via exponents θ_J and θ_Q . While θ_J and θ_Q are completely defined by the dynamical model chosen, their role changes depending on whether contributions from neighbouring nodes are important ($\theta_Q > 0$) or vanish in the large degree limits ($\theta_Q < 0$). The authors then θ_Q to prototypical models in the literature and proceed to show that using the local results it is possible to investigate the role of node degrees and higher-order motifs at the global level, that is, along trajectories of arbitrary length.

In particular, they show that both the mean and variance of the degree along a trajectory affect the propagation time of a perturbation. In particular, higher/lower degree means/variances can increase/decrease response time depending on the sign of θ_J .

I think overall the paper is an important contribution and should be considered for publication in Nature Communications. However, I believe that the current form of the manuscript could be improved to make the paper more readable and amenable to a wider community. I also found some discrepancies/errors which should be addressed.

Major comments:

- I think the authors should consider changing the term motifs to loops in the title and in the text. While I agree that closed paths are indeed motifs, there are many more (connected) motifs of sizes 3,4,5 than only closed paths. Using high-order motifs seems confusing/misleading, also in consideration of the current confusion about the use of higher-order for temporal paths versus polyadic interactions, etc.

- the paper is quite dense. This is not a problem in itself, but I believe that the main text could be significantly expanded in terms of explanation and examples to aid the reader's comprehension while staying within Nat Comms' limits.

It is clear that the authors did a lot of work (the SI is clear testimony to that), but I have a feeling that the introduction and discussion could be expanded to cover the importance of the results.

In the section about the local propagation, the exponent θ_Q appears quite suddenly in eq 8 and so does θ_J in Eq. 10. I would suggest expanding the discussion and introduction of these quantities when they appear, possibly better relating them to their interpretation. I am aware that reading through the SI it is possible to get a sense of this, but it's also a lot of material that most readers won't have time/expertise to sort through.

- in the section about the local propagation the authors mention mutualistic and biochemical dynamics as dynamics with $\theta_Q > 0$, but in Table 1 and in the SI they show them to have $\theta_Q < 0$. Is this just a typo or am I missing something deeper?

- the authors claim that signal propagation is increasingly limited with clustering coefficient and use Fig.4 to show it, but in Fig.4c the distributions look very similar to each other by eye. Are they significantly different (and under what test/lens)?

Reviewer #2 (Remarks to the Author):

In their manuscript "Impact of higher-order network motifs on response dynamics" the Authors treat the effect of triangles and higher-order motifs on system dynamics. In particular they have introduced two topology-independent exponents in order to quantify the impact of the motifs, and they develop a theoretical framework for different dynamical behaviors, from human to epidemic and so on.

The Authors mainly study the behavior of propagation time from a local perspective, thus revealing the effects of triangles and independent edges, and then they extend the results by quantifying the propagation time from a global perspective.

The manuscript is well written and represents a strong step forward in terms of the theoretical treatment of the effects that nodes along propagation paths induce, particularly given the strong interest in higher-order interactions that currently resides in the network science community. I believe that the paper will be of significant interest to a wide audience of researchers working on the interplay between topological properties and interaction dynamics.

The whole paper is well organized, however the explanation of the two scaling exponents should be more accurate.

Most of the informations proposed in Section 2.3 of the Supplemental material should be moved (part of this section, not all of it) into the main text.

In my opinion the fact that the propagation time is affected at the same time by nodal dynamics, by the independent edges and by the triangles is very important, maybe is the crucial point of the paper, and the I would prefer a more deep discussion on this.

For example, also Table S1 contains interesting informations, clearly shows the importance of each motif when the target node receives a perturbation. It is not obvious that the role of line and triangles is more crucial with respect to other motifs such as polygon with more than 3 edges. I think that this should be mentioned in the main text.

Reviewer #3 (Remarks to the Author):

The manuscript "Impact of higher-order network motifs on response dynamics" reports an extensive work on the effect of triangles in a network dynamics. In particular the authors consider a general dynamical system, composed by self-dynamics and nodes interaction, which admits at least a fixed point. They hence perturb one node state and observe how the perturbation propagates to the other nodes (from local and global point of view) making the entire system change its state. They observe a general tendency to slow the dynamics due to triangles.

The work is interesting and I think it adds a brick to the field of dynamics on higher-order networks. The article is clear and well written.

I only have some doubts that require a clarification and some small changes to the article that would improve its presentation.

First of all, the title seems a bit too generic for this work. I would rather suggest “Impact of higher-order network motifs on dynamical response to perturbations”, or similar.

Moreover, since the title mentions higher-order networks, but then only the impact of triangles is explored, I would be curious to see (even not analytically but just numerically) if also higher orders, like tetrahedra, have an impact.

Another doubt that I have concerns the chosen dynamical system and how much the obtained results depend on the fixed point stability. If I understood correctly, the system has to change state with the perturbation, but that means that the perturbation has to be big enough to move the state from its original basin of attraction. This provides a constraint on the initial perturbation, is it correct?

Fig. 1: it is a clear presentation of the idea. However I think that the chosen network is not really representative of real networks. In ER networks indeed we usually do not have a high clustering. Triangles are found here just because the networks are very dense. I suggest to provide an example on a real network first in order to give the message that triangles are usually present in real-world. Then the synthetic example could be done on a more realistic network (also characterized by high clustering) like Watts-Strogatz networks.

From fig. 1(d) we observe that nodes that are part of triangles are slower in reaching the new state. However in the toy example of fig. 1(f) it seems that node *i* is the slowest, which seems in contrast with the previous result. Am I missing something?

Fig. 4(a): we do not well recognize purple nodes from blue ones, I suggest to change colors or node sizes.

Figs 4(b) and (c): I wonder why the range of number of triangles is so different (between 2 and 6 in (b) and between 478 and 3647 in (c)). Also, the slowing effect at increasing the number of triangles does not appear significant from these examples (same in S11 and S12). Would it make sense to try with another example of dynamics?

Response to Reviewers

Dear Editor,

We are very thankful for the positive attitude of all three referees toward our manuscript and would like to thank the referees for their thorough assessments. We have changed the manuscript and the supplementary material accordingly, with changes highlighted in blue.

Here, we address the raised criticism, to which we now respond on a point-by-point basis.

REVIEWER COMMENTS WITH OUR REPLIES

Reviewer #1 (Remarks to the Author):

The manuscript "Impact of higher-order network motifs on response dynamics" focuses on the effects that the presence of triangles and other types of higher-order motifs have on the characteristic time over which a perturbation propagates through a network. The authors in particular focus on specific type of motifs, that is, closed paths of variables length (edge, triangle, square, pentagon, etc), and derive general expressions valid for many different types of models (ranging from epidemic to biochemical ones).

The central result is --at the level of local perturbation-- a direct link between the response time of a unit i with its degree via exponents θ_J and θ_Q . While θ_J and θ_Q are completely defined by the dynamical model chosen, their role changes depending on whether contributions from neighbouring nodes are important ($\theta_Q > 0$) or vanish in the large degree limits ($\theta_Q < 0$). The authors then θ_Q to prototypical models in the literature and proceed to show that using the local results it is possible to investigate the role of node degrees and higher-order motifs at the global level, that is, along trajectories of arbitrary length. In particular, they show that both the mean and variance of the degree along a trajectory affect the propagation time of a perturbation. In particular, higher/lower degree means/variances can increase/decrease response time depending on the sign of θ_J .

I think overall the paper is an important contribution and should be considered for publication in Nature Communications. However, I believe that the current form of the manuscript could be improved to make the paper more readable and amenable to a wider community. I also found some discrepancies/errors which should be addressed.

Reply: We highly appreciate your positive assessment together with the very instructive comments and suggestions you made. Thank you! We have modified the manuscript accordingly.

Major comments:

- I think the authors should consider changing the term motifs to loops in the title and in the text. While I agree that closed paths are indeed motifs, there are many more (connected) motifs of sizes 3,4,5 than only closed paths. Using high-order motifs seems confusing/misleading, also in consideration of the current confusion about the use of higher-order for temporal paths versus polyadic interactions, etc.

Reply: The current theoretical framework, for local and global response dynamics, focuses on the impact "independent edges" and triangles have (being both basic undirected motifs). Numerically, we also evaluate the impact of squares, pentagons and now tetrahedrons on the response dynamics (cf. new Tables S1 and S2). We have emphasized the difference of different motif types and changed the introduction accordingly. We have renamed the title of our work to "Impact of basic network

motifs on the collective response to perturbations". We think that the overall notion of "basic motifs" as now used throughout the manuscript does make sense and can be deemed agreeable. In this context, we have completely removed subphrases with "higher-order" as it can be indeed confused with, e.g., polyadic interactions. Thank you for this valuable input!

- the paper is quite dense. This is not a problem in itself, but I believe that the main text could be significantly expanded in terms of explanation and examples to aid the reader's comprehension while staying within Nat Comms' limits.

It is clear that the authors did a lot of work (the SI is clear testimony to that), but I have a feeling that the introduction and discussion could be expanded to cover the importance of the results.

Reply: Thanks for raising this, we fully agree. Following your suggestions, we have expanded *Introduction*, *Results*, and *Discussion* accordingly. We have also shorted the *Abstract* to stay within word limits. In the *Introduction*, we now specify the importance of motifs on system dynamics with an additional paragraph. For the *Results* section, we tried as much as possible not to refer to the SM but rather explain details on the analytical framework, and give the main idea of the given derivation. In the *Discussion*, we have included the discussion of the current theoretical framework's limitation and a brief outlook, as also suggested by other referees.

Additionally, in the supplementary material, we worked out two parts:

1) Quantification of the impact of response dynamics from various basic motifs (including edges, triangles, squares, pentagons, and tetrahedrons), as summarized in Tables S1 and S2.

2) Quantification the impact of response dynamics of more complex network settings, including both independent edges and triangles, as shown in Figure S10.

In the section about the local propagation, the exponent θ_Q appears quite suddenly in eq 8 and so does θ_J in Eq. 10. I would suggest expanding the discussion and introduction of these quantities when they appear, possibly better relating them to their interpretation. I am aware that reading through the SI it is possible to get a sense of this, but it's also a lot of material that most readers won't have time/expertise to sort through.

Reply: Following your suggestions, we have substantially expanded *Introduction*, *Results*, and *Discussion* accordingly, in particular the manuscript between Eqs. (7) and (13) on the scaling exponents θ_Q and θ_J and their interpretation and connection.

- in the section about the local propagation the authors mention mutualistic and biochemical dynamics as dynamics with $\theta_Q > 0$, but in Table 1 and in the SI they show them to have $\theta_Q < 0$. Is this just a typo or am I missing something deeper?

Reply: Thank you very much for the careful reading and for pointing out this typo! Indeed, mutualistic and biochemical dynamics are both characterized by $\theta_Q < 0$, as shown in Table 1. We have corrected for this typo. It read now "...we find that regulatory, human, mutualistic, biochemical and epidemics dynamics are characterized by $\theta_Q < 0$, whereas population and inhibitory neuronal dynamics show $\theta_Q > 0$ ".

- the authors claim that signal propagation is increasingly limited with clustering coefficient and use

Fig.4 to show it, but in Fig.4c the distributions look very similar to each other by eye. Are they significantly different (and under what test/lens)?

Reply: Figure 4 shows the global signal propagation across the whole network, and the signal propagates from the source node m to remote nodes across various layers and pathways. With increasing clustering coefficient, the response time does significantly increase, but not drastically.

First of all, not only local topological properties, but also the degree sequence and the length of pathways play a crucial role for the response time. Second, for a given pathway, not all triangles of the entire network contribute (or slow) to the propagation, but mainly those along the path.

For the sake of clarity, we now study the propagation as a function of the clustering coefficient in Fig. 4. Overall, we now believe to provide a coherent picture.

Reviewer #2 (Remarks to the Author):

In their manuscript "Impact of higher-order network motifs on response dynamics" the Authors treat the effect of triangles and higher-order motifs on system dynamics. In particular they have introduced two topology-independent exponents in order to quantify the impact of the motifs, and they develop a theoretical framework for different dynamical behaviors, from human to epidemic and so on.

The Authors mainly study the behavior of propagation time from a local perspective, thus revealing the effects of triangles and independent edges, and then they extend the results by quantifying the propagation time from a global perspective.

The manuscript is well written and represents a strong step forward in terms of the theoretical treatment of the effects that nodes along propagation paths induce, particularly given the strong interest in higher-order interactions that currently resides in the network science community. I believe that the paper will be of significant interest to a wide audience of researchers working on the interplay between topological properties and interaction dynamics.

Reply: We highly appreciate your positive attitude toward our manuscript and your very helpful comments and suggestions.

The whole paper is well organized, however the explanation of the two scaling exponents should be more accurate.

Reply: We have substantially expanded the discussion, in particular on the scaling exponents θ_Q and θ_J and their interpretation and connection, see text in the manuscript between Eqs. (7) and (13), highlighted in blue.

Most of the informations proposed in Section 2.3 of the Supplemental material should be moved (part of this section, not all of it) into the main text.

Reply: We fully agree and have revised to large extent the Results section.

In my opinion the fact that the propagation time is affected at the same time by nodal dynamics, by the independent edges and by the triangles is very important, maybe is the crucial point of the paper, and the I would prefer a more deep discussion on this.

Reply: We fully agree and have expanded the manuscript accordingly. Following your suggestions, in the current version of the supplementary material, we now study the more complex case of coexisting independent edges and triangles, as shown in Fig. S10.

For example, also Table S1 contains interesting informations, clearly shows the importance of each motif when the target node receives a perturbation. It is not obvious that the role of line and triangles is more crucial with respect to other motifs such as polygon with more than 3 edges. I think that this should be mentioned in the main text.

Reply: Thank you very much for this advice! The crucial roles of edges and triangles were not illustrated clearly in the manuscript. To address this, we additionally quantify and study the impact of response dynamics for various motifs, including edges, triangles, squares, pentagons, and tetrahedrons, as shown in Tables S1 and S2 of the revised supplementary material. In particular, the quantification integrates the response dynamics difference before and after cutting signal flows across motifs. The numerical results indicate that the impacts of n-gons ($n \geq 3$) decrease with increasing n, from triangles to Pentagon. Also, the effects of triangles and of the Tetrahedron are very similar.

Reviewer #3 (Remarks to the Author):

The manuscript “Impact of higher-order network motifs on response dynamics” reports an extensive work on the effect of triangles in a network dynamics. In particular the authors consider a general dynamical system, composed by self-dynamics and nodes interaction, which admits at least a fixed point. They hence perturb one node state and observe how the perturbation propagates to the other nodes (from local and global point of view) making the entire system change its state. They observe a general tendency to slow the dynamics due to triangles.

The work is interesting and I think it adds a brick to the field of dynamics on higher-order networks. The article is clear and well written.

I only have some doubts that require a clarification and some small changes to the article that would improve its presentation.

Reply: We genuinely appreciate your helpful comments and suggestions and positive attitude toward our manuscript. Thank you!

First of all, the title seems a bit too generic for this work. I would rather suggest “Impact of higher-order network motifs on dynamical response to perturbations”, or similar.

Reply: We agree that the title was a bit too generic for this work.

We have renamed the title to “*Impact of basic network motifs on the collective response to perturbations*”. We think that the overall notion of “basic motifs” as changed throughout the manuscript does make sense now. In this context, we have completely removed subphrases with “higher-order”, to avoid any confusion.

Moreover, since the title mentions higher-order networks, but then only the impact of triangles is explored, I would be curious to see (even not analytically but just numerically) if also higher orders, like tetrahedra, have an impact.

Reply: Thanks for your suggestion. We have conducted an experiment to explore the impact of higher-order motifs such as the ones shown in Table S2 in the supplementary material. In particular, we study a network including two triangles and one tetrahedra, and provide the quantification of the impact of the response dynamics of various motifs. The effects of triangles and tetrahedra are very similar with respect to the response times.

Another doubt that I have concerns the chosen dynamical system and how much the obtained results depend on the fixed point stability. If I understood correctly, the system has to change state with the perturbation, but that means that the perturbation has to be big enough to move the state from its original basin of attraction. This provides a constraint on the initial perturbation, is it correct?

Reply: Thank you for raising this important point! We agree that for large perturbations, the system may move from the original collective state to a new state, which cannot be predicted by linear response theory. In the revised manuscript we have highlighted that we study a *small* permanent perturbation driving the system from the original state away to a new collective stationary state. In particular, we close the *Discussion* section with a suitable discussion on the limitations (and outlook) of our developed framework.

Fig. 1: it is a clear presentation of the idea. However I think that the chosen network is not really representative of real networks. In ER networks indeed we usually do not have a high clustering. Triangles are found here just because the networks are very dense. I suggest to provide an example on a real network first in order to give the message that triangles are usually present in real-world. Then the synthetic example could be done on a more realistic network (also characterized by high clustering) like Watts-Strogatz networks.

Reply: Thank you very much for this excellent suggestion, we fully agree. As a result, we have amended Figure 1 (a-c) in the revision as follows:

(a) Now showing a schematic plot of a protein-protein network, a small-world network, and an Erdős-Renyi (ER) network.

(b) Share of motifs (edges, triangles, squares, pentagons) for each network.

(c) Share of edges as part of triangles, squares and pentagons.

From fig. 1(d) we observe that nodes that are part of triangles are slower in reaching the new state. However in the toy example of fig. 1(f) it seems that node *i* is the slowest, which seems in contrast with the previous result. Am I missing something?

Reply: Thanks a lot for this comment! The two panels in Fig. 1 convey different messages. Fig 1(d) is the result of the ER network with 100 nodes in Fig 1(a), whereas Fig 1(f) depicts the situation of the shown 4-node network, to justify our decomposition of networks into independent edges and triangles. We apologize for the confusion. Fig 1(d) demonstrates that, on average, target nodes “*i*” in (independent) edges respond faster than target nodes as part of triangles, given they have the same distance to the source.

In contrast, In Fig. 1(f) we focus on three nodes of the simple 4-node-demonstration network. Note that the j-node responds slower, but also note that it is two hops apart from source m, and not only one hop as the nodes i and h.

Fig.1 (f) should demonstrate two observations:

The convergence behavior of responses $\Delta x_j(t)$ and $\Delta x_h(t)$ are on the same time scale as $\Delta x_i(t)$. In particular, all responses are non-instantaneous, which is conflicting to a non-vanishing $E_{im}(t)$, Eq.6. This demonstrates that, even for such a simple case, the implicit assumption made in existing theory ($E_{im}(t)=0$), is problematic.

Second, the two responses, $\Delta x_j(t)$ and $\Delta x_h(t)$, of the nodes directly connected to the target node i, differ from each other, which our framework accounts for – also in contrast to existing theory.

These observations have prompted us to decompose networks into independent edges and triangles for our overall analysis. We have changed the main text in the manuscript accordingly to clarify this and added the supplementary Tables S1 & S2.

To clarify that nodes in independent edges respond faster than nodes as part of triangles, we now present Tables S1 & S2 in the supplementary material, where we cut one direction of the signal propagation to show that target nodes in (independent) edges respond faster than target nodes as part of triangles, given they have the same “effective” distance to the source.

In addition, we emphasize in the manuscript that an independent edge is an edge not directly connected to the source, hence it is at least two hops apart from the source.

Fig. 4(a):we do not well recognize purple nodes from blue ones, I suggest to change colors or node sizes.

Reply: Thanks for this suggestion. We have changed the color of nodes to “brown” to highlight the spreading process in the manuscript and the corresponding figures in the supplementary material.

Figs 4(b) and (c): I wonder why the range of number of triangles is so different (between 2 and 6 in (b) and between 478 and 3647 in (c)). Also, the slowing effect at increasing the number of triangles does not appear significant from these examples (same in S11 and S12). Would it make sense to try with another example of dynamics?

Reply: Thank you for raising this. Please find our response as made earlier to Reviewer #1:

Figure 4 shows the global signal propagation across the whole network, and the signal propagates from the source node m to remote nodes across various layers and pathways. With increasing clustering coefficient, the response time does significantly increase, but not drastically.

First of all, not only local topological properties, but also the degree sequence and the length of pathways play a crucial role for the response time. Second, for a given pathway, not all triangles of the entire network contribute (or slow) to the propagation, but mainly those along the path.

For the sake of clarity, we now study the propagation as a function of the clustering coefficient in Fig. 4. Overall, we now believe to provide a coherent picture.

REVIEWERS' COMMENTS

Reviewer #2 (Remarks to the Author):

Reading the new version of the paper I'm very satisfied of the revision. Now the crucial point of the paper and studied by the Authors is more clear and also the overall presentation of the results have benefit from the revision.

I recommend the paper for publication.

Reviewer #3 (Remarks to the Author):

The authors have addressed all of my previous comments and the article has been heavily modified and improved. I suggest acceptance on Nature Communications.